# Epidemiological, clinical, and geographical characterization of Leprosy in the County of Santarém-Pará: Insights for effective control and targeted intervention

Edson Jandrey Cota Queiroz[1©], Ingrid Nunes da Rocha[1©], Lívia de Aguiar Valentim[1©]*, Thiago Junio Costa Quaresma[2‡], Zilmar Augusto de Souza Filho[3‡], Sheyla Mara Silva de Oliveira[1‡], Franciane de Paula Fernandes[1‡], Caroline Gomes Macedo[1©], Tatiane Costa Quaresma[1©], Waldiney Pires Moraes[2‡]

1 Department Health at the University of the State of Pará, Belém, Brazil, 2 Department of Health at the Federal University of Western Pará, Belém, Brazil, 3 Department of Health at the Federal University of Amazonas, Manaus, Brazil

© These authors contributed equally to this work.
‡ TJCQ, ZASF, SMSO, FPF and WPM also contributed equally to this work.
* livia.valentim@uepa.br

**Data Availability Statement:** The data analyzed in this study are not publicly available due to privacy

## Abstract

Leprosy is an infectious disease characterized by slow and chronic evolution, caused by *Mycobacterium leprae* and or *Mycobacterium lepromatosis*, an intracellular alcohol-acid-resistant (BAAR) bacillus. The objective of this study was to provide an epidemiological, clinical, and geographic characterization of leprosy in the city of Santarém-Pará during the period 2011–2020. A cross-sectional, descriptive, and quantitative approach was used, employing maps and tables to illustrate clinical and epidemiological variables, including: sex, age, race, area of residence, operational classification, clinical form, number of skin lesions, number of affected nerves, and health units. During the analyzed period, 581 cases of leprosy were diagnosed, resulting in the following cumulative incidence rates: male (60%); age over 15 years (94%); urban area (73%); multibacillary (74%); borderline form (46%); skin lesions greater than 5 (34%); and no nerves affected (68%). In the urban perimeter, a higher cumulative incidence of cases was observed in the central area with 133 cases. However, the health unit reporting the largest number of cases belonged to the southern area, specifically the Basic Health Unit of Nova República, with 48 cases. This study highlights the need to characterize the nuances of leprosy and its variability within the urban environment, according to different areas. Further research is essential to inform the implementation of public policies aimed at addressing the population with the highest vulnerability index, thereby reducing leprosy rates in Santarém.

and ethical restrictions. The dataset used for this research contains sensitive patient information and is subject to confidentiality agreements. Therefore, the data cannot be shared openly. However, researchers interested in accessing the data can contact the ethics committee, e-mail: ceptapajos@uepa.br.

**Funding:** The author(s) received no specific funding for this work.

**Competing interests:** The authors have declared that no competing interests exist.

## Author summary

This article addresses the incidence of leprosy in Santarém, Brazil, during the COVID-19 pandemic. In 2020, there was a 41% reduction in cases compared to the average between 2015 and 2019, possibly due to the cancellation and rescheduling of appointments, affecting diagnoses and follow-up. Men constituted 60% of the cases, with their hesitation to seek care associated with late diagnoses. Alarmingly, 6% of patients were under 14 years old, indicating possible intrafamilial transmission. Socioeconomic and ethnic inequalities were evident, with 74% of cases in individuals of mixed race. The article emphasizes the importance of addressing leprosy not only as a public health challenge but also as a reflection of social and economic inequalities in Brazil. It proposes comprehensive health policies, expansion of medical services in remote regions, and educational initiatives for awareness and early diagnosis, especially in vulnerable communities. The association between age and clinical form of the disease, as well as spatial distribution analyses, provides crucial insights for developing targeted prevention and treatment strategies.

## Introduction

Leprosy remains a significant public health concern in several regions worldwide, including Brazil [1,2]. It is an infectious and contagious disease with slow and chronic progression, caused by *Mycobacterium leprae* and or *Mycobacterium lepromatosis* [3–5]. The etiological agent is an intracellular acid-fast bacillus that affects Schwann cells and the skin, particular Remak Schwann cells, leading to dermatoneurological clinical manifestations and irreversible physical deformities if left untreated [6,7]. The disease's low virulence and slow multiplication result in chronic development, affecting both the skin and nerves [8–10]. Clinically, it presents through dermatoneural involvement, falling within two stable clinical spectrums (tuberculoid and lepromatous poles), with unstable forms between them [10].

Patients with at least one of the World Health Organization 's cardinal signs will have their diagnosis made based on clinical findings alone [11]. Timely diagnosis through physical examination or bacilloscopy of dermal scrapings and the use of multidrug therapy (rifampicin, dapsone, and clofazimine) can prevent disease progression and disability [12]. Commonly known as "Hansen's disease" or "leprosy," this condition has faced intense stigmatization throughout history due to the physical deformities it can cause, leading to social isolation for affected individuals [13]. For treatment purposes, patients are classified as paucibacillary (PB) or multibacillary (MB).

The primary objectives of leprosy treatment are disease cure for paucibacillary patients, prevention of physical disabilities, and management of leprosy reactions. The World Health Organization's operational classification [14] divides patients into paucibacillary (PB) and multibacillary (MB), with PB presenting 1 to 5 skin lesions without acid-fast bacilli in histopathological examination or bacilloscopy. MB forms exhibit more than five skin lesions, nerve involvement, or bacilli present in smear/biopsy regardless of the number of skin lesions [15]. Unfortunately, the accuracy of leprosy diagnosis and classification by healthcare professionals, combined with limited availability of bacilloscopy, has led to misdiagnoses, with some MB patients being treated as PB. To prevent such occurrences, the WHO [14] recommends a therapeutic scheme with three drugs (Rifampicin + Clofazimine + Dapsone) for all patients, lasting 6 months for PB cases and 12 months for MB cases.

Effective disease control demands a comprehensive approach that goes beyond medical treatment. Social stigmatization is a significant barrier faced by individuals affected by leprosy

[16–18]. The physical deformities resulting from the disease can lead to social exclusion, discrimination, and prejudice, significantly affecting the patients' quality of life [19]. To combat this stigmatization and improve patient inclusion, public awareness of the disease is essential, emphasizing its treatable nature and advancements in treatment. Integrated education and awareness campaigns can help dispel myths surrounding leprosy and challenge negative beliefs and attitudes; [16,20].

According to the WHO [14], 202,185 new cases were reported worldwide, with Brazil contributing 27,863 of these cases, making it a country with a high burden of disease and ranking second in number of cases, behind only India [21]. To address this, the Brazilian Ministry of Health created the National Strategy for Leprosy Confrontation (2019–2020) with the goal of reducing the disease burden in the country by the end of 2022. To achieve this, the strategy relies on Primary Health Care (PHC) through the Family Health Strategy (FHS), implementing a set of actions developed by a multi-professional FHS team focused on disease control, including the notification of confirmed cases, distribution of medications, and administering the BCG (Bacillus Calmette-Guérin) vaccine [12].

Regarding the treatment of leprosy patients co-infected with HIV, studies highlight complexities and challenges. It was observed that infection with this virus does not seem to change the incidence or clinical spectrum of the disease, and does not cause changes, or does so discreetly, in the course of leprosy [22,23]. Therefore, the symptoms, reactions and impact of ART and PREP are uncertain in cases of HIV/Leprosy, making it difficult for the patient to adhere to available treatment [24].

In this context, understanding the location of health teams' activities is crucial for effective planning and execution of actions for the population's well-being. The Family Health Program (FHP) plays a fundamental role in registering families linked to the Health Unit, providing essential data for territorialization consolidation. Geographic Information Systems (GIS) are used for the electronic processing of geographically referenced data, allowing for the storage and manipulation of this information [25].

Considering that leprosy is endemic in various regions of Brazil, strengthening the capacity of health services to ensure early and accurate diagnosis is crucial [26]. Training programs for healthcare professionals should be implemented, emphasizing the importance of clinical suspicion and bacilloscopy to identify the disease's different clinical spectrums [27,28]. Furthermore, professionals' access to technological resources for monitoring cases, encouraging diagnoses, such as bacilloscopy, should be expanded in areas with a high cumulative incidence of leprosy, ensuring reliable diagnosis and adequate classification of patients as paucibacillary or multibacillary [29]. These measures can help prevent misdiagnoses and ensure that patients receive appropriate treatment according to the WHO's operational classification.

By adopting multidisciplinary and collaborative approaches, it is possible to strengthen leprosy control and improve the quality of life for patients with the disease. Combating stigmatization, raising public awareness, and enhancing diagnostic and treatment capabilities can help in the efforts to eliminate leprosy as a public health problem in Brazil and other parts of the world. The epidemiological, clinical, and geographical characterization of leprosy in the city of Santarém-Pará between 2011 and 2020, as presented in this study, can provide valuable insights for formulating more effective strategies for prevention, control, and targeted intervention in this specific region.

## Methodology

### Ethics statement

The study was approved, opinion 4.470.346, by the Research Ethics Committee (CEP) of the State University of Pará, written informed consent was obtained from all participants, including parents and/or guardians for children, and all methods were conducted in accordance with current guidelines and standards.

### Leprosy research approach and mapping methodology

This research follows an analytical cross-sectional study design, employing a descriptive and quantitative approach for the clinical-epidemiological characterization of leprosy cases in the city of Santarém, Pará. Georeferencing tools were used to correlate the number of cases with notifying units within zones of the urban perimeter, allowing for mapping.

Santarém is located in the western part of Pará and has a population of 294,280 inhabitants spread across 17,898.388 km$^2$, according to the latest census by the Brazilian Institute of Geography and Statistics - IBGE [30], with the majority residing in the urban zone, representing 215,790 (73.25%) of the total population.

Data on leprosy cases between 2011 and 2020 were obtained from the database of the Municipal Health Department (SEMSA) of Santarém-PA. The data were categorized into epidemiological, clinical, and cases by notifying units variables.

The epidemiological variables included gender, age, skin color, and residential zone of the diagnosed cases. The clinical variables encompassed the operational classification, clinical form, number of skin lesions, and number of affected nerves.

Notifying units were ranked based on the number of registered cases. Electronic searches for secondary data were conducted on the website of the Municipal Government of Santarém to find the addresses of the identified units. This information was essential for differentiating each Basic Health Unit according to its zoning within the urban perimeter, as recommended in the Municipal Master Plan of Santarém. Shapefiles, equipped with geographic information provided by IBGE. In conducting the spatial analysis in this study, we utilized QGIS 3.34 software for the creation and presentation of maps. The geographic layer pertaining to the population and neighborhood boundaries was sourced from data provided by the Brazilian Institute of Geography and Statistics (IBGE).

In our initial presentation of the spatial risk assessment in Fig 1, we utilized two distinct approaches to understand the distribution of risk across the studied region. The first approach was a neighborhood-specific risk calculation, which provided a detailed, localized understanding of risk levels. The second was a spatial clustering tool, designed to identify broader patterns of risk across multiple neighborhoods. It is noteworthy that the generated map represented the analysis of relative risk and clusters, which were calculated using the Statistical Package for the Social Sciences (SPSS) software. This methodological approach allowed for a more comprehensive understanding of the spatial dynamics of the disease, contributing to the identification of areas requiring targeted interventions and specific control strategies.

Microsoft Office Word version 2019 was used for textual production. While the table of clinical-epidemiological variables contains all records made in the city, the table of cases by notifying unit includes only diagnoses within the urban perimeter, excluding expansion and rural areas. Descriptive statistics were employed, presenting absolute and relative frequencies, along with inferential analysis through the Chi-Square Test using SPSS Statistics 20 software. Percentages were obtained using Microsoft Office Excel version 2019. Notably, the p-value was

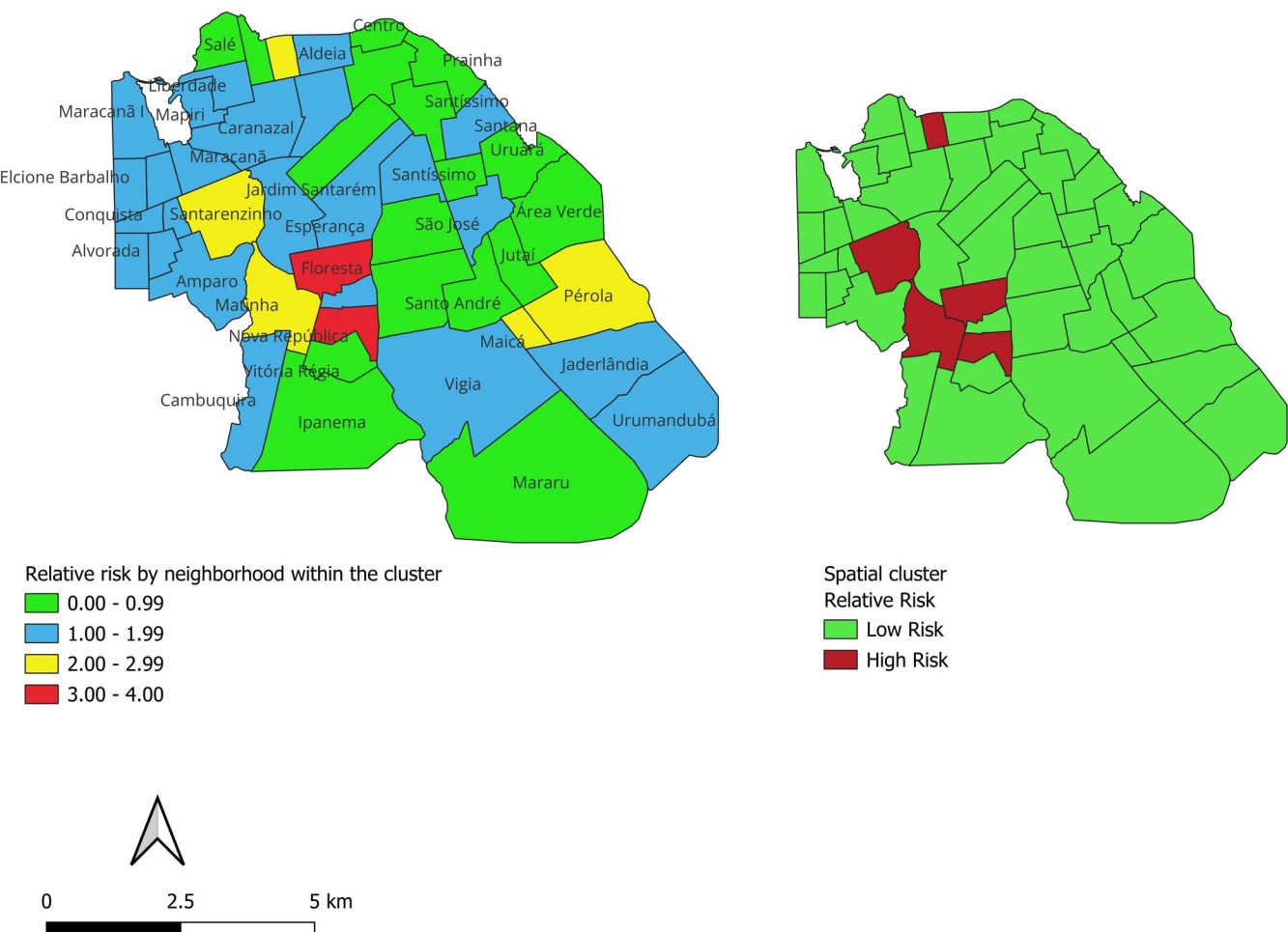

**Fig 1. Spatial clusters and relative risk of Leprosy detection in Santarém.** Source: Map built by the authors in QGIS, from the shapefile obtained by Instituto Brasileiro de Geografia e Estatística–IBGE, available at the link https://www.ibge.gov.br/geociencias/downloads-geociencias.html?caminho=cartas_e_mapas/mapas_municipais/colecao_de_mapas_municipais/2020/PA/.

not presented for data related to the year 2020 due to the reduced number of registered cases that year.

## Results

Between 2011 and 2020, a total of 581 new cases of leprosy were reported in the city of Santarém. Tables 1 and 2 present data related to the epidemiology and clinical aspects of the disease in Santarém, Pará. The year 2014 recorded the highest number of leprosy cases, with 83 cases, while the lowest number of cases was observed in 2020, with 24 cases, possibly influenced by the COVID-19 pandemic.

Concerning the clinical forms presented at the time of diagnosis, as shown in Table 2, it was found that among the affected patients, Dimorphous Leprosy (DL) and Lepromatous leprosy (LL) were responsible for the majority of cases, with 46% and 24%, respectively, totaling 70%. In contrast, Indeterminate Leprosy (IL) and Tuberculoid Leprosy (TL) accounted for 13% and 13%, respectively. The inclusion of Ridley-Jopling classifications and operational classification in Table 2 is of great importance in the analysis of leprosy in Santarém. These classifications provide a comprehensive view of the disease, considering both its clinical and operational

**Table 1. Epidemiological Characterization of Leprosy Cases According to Year of Notification (2011–2020) in Santarém City.**

| VARIABLES EPIDEMIOLOGICAL | 2011 | | 2012 | | 2013 | | 2014 | | 2015 | | 2016 | | 2017 | | 2018 | | 2019 | | 2020 | | TOTAL | |
|---|---|---|---|---|---|---|---|---|---|---|---|---|---|---|---|---|---|---|---|---|---|---|
| | n | % | n | % | n | % | n | % | n | % | n | % | n | % | n | % | n | % | n | % | n | % |
| **Sex** | | | | | | | | | | | | | | | | | | | | | | |
| Masculine | 43 | 61 | 40 | 65 | 34 | 60 | 45 | 55 | 40 | 63 | 41 | 61 | 34 | 67 | 30 | 58 | 33 | 65 | 11 | 46 | 351 | 60 |
| Feminine | 28 | 39 | 22 | 35 | 23 | 40 | 37 | 45 | 24 | 38 | 26 | 39 | 17 | 33 | 22 | 42 | 18 | 35 | 13 | 54 | 230 | 40 |
| *p-value* | *0.0966* | | *0.0309* | | *0.185* | | *0.439* | | *0.0608* | | *0.0872* | | *0.025* | | *0.331* | | *0.049* | | - | | - | |
| **Age** | | | | | | | | | | | | | | | | | | | | | | |
| 0–14 years | 4 | 6 | 2 | 3 | 2 | 4 | 9 | 11 | 2 | 3 | 6 | 9 | 2 | 4 | 0 | 0 | 1 | 4 | 4 | 17 | 32 | 6 |
| 15 or more | 67 | 94 | 60 | 97 | 54 | 96 | 74 | 89 | 63 | 97 | 60 | 91 | 49 | 96 | 52 | 100 | 50 | 96 | 20 | 83 | 549 | 94 |
| *p-value* | <0.001 | | <0.001 | | <0.001 | | <0.001 | | <0.001 | | <0.001 | | <0.001 | | <0.001 | | <0.001 | | - | | - | |
| **Skin color** | | | | | | | | | | | | | | | | | | | | | | |
| Not filled | 0 | 0 | 0 | 0 | 1 | 2 | 0 | 0 | 0 | 0 | 1 | 2 | 2 | 4 | 0 | 0 | 0 | 0 | 0 | 0 | 4 | 1 |
| White | 10 | 14 | 7 | 11 | 7 | 12 | 14 | 17 | 10 | 16 | 7 | 10 | 6 | 12 | 8 | 15 | 3 | 12 | 2 | 8 | 74 | 13 |
| Black | 12 | 17 | 7 | 11 | 10 | 18 | 8 | 10 | 8 | 13 | 3 | 4 | 7 | 14 | 2 | 4 | 4 | 14 | 6 | 25 | 67 | 11,5 |
| Yellow | 0 | 0 | 1 | 2 | 0 | 0 | 1 | 1 | 0 | 0 | 0 | 0 | 1 | 2 | 1 | 2 | 0 | 0 | 10 | 42 | 14 | 2 |
| Brown | 49 | 69 | 46 | 74 | 39 | 68 | 58 | 72 | 46 | 72 | 57 | 84 | 35 | 69 | 39 | 75 | 44 | 69 | 6 | 25 | 419 | 72 |
| Indigenous | 0 | 0 | 1 | 2 | 0 | 0 | 0 | 0 | 0 | 0 | 0 | 0 | 0 | 0 | 2 | 4 | 0 | 0 | 0 | 0 | 3 | 0,5 |
| *p-value* | <0.001 | | <0.001 | | <0.001 | | <0.001 | | <0.001 | | <0.001 | | <0.001 | | <0.001 | | <0.001 | | - | | - | |
| **Zone in residence** | | | | | | | | | | | | | | | | | | | | | | |
| Not filled | 2 | 3 | 2 | 3 | 17 | 30 | 1 | 1 | 1 | 2 | 0 | 0 | 2 | 4 | 0 | 0 | 0 | 0 | 0 | 0 | 25 | 4 |
| Urban | 49 | 69 | 50 | 81 | 27 | 48 | 68 | 82 | 51 | 78 | 54 | 82 | 34 | 67 | 35 | 67 | 41 | 80 | 14 | 58 | 423 | 73 |
| Rural | 20 | 28 | 10 | 16 | 11 | 20 | 12 | 14 | 11 | 17 | 11 | 17 | 14 | 27 | 17 | 33 | 10 | 20 | 8 | 33 | 124 | 21 |
| Periurban | 0 | 0 | 0 | 0 | 1 | 2 | 2 | 2 | 2 | 3 | 1 | 2 | 1 | 2 | 0 | 0 | 0 | 0 | 2 | 8 | 9 | 2 |
| *p-value* | <0.001 | | <0.001 | | <0.001 | | <0.001 | | <0.001 | | <0.001 | | <0.001 | | <0.001 | | <0.001 | | - | | - | |
| **TOTAL** | **71** | **100** | **62** | **100** | **56** | **100** | **83** | **100** | **65** | **100** | **66** | **100** | **51** | **100** | **52** | **100** | **51** | **100** | **24** | **100** | **581** | **100** |

Source: Secretary City of Health in Santarem (SEMSA), Santarém/PA, 2021.

aspects. The Ridley-Jopling classification, with its clinical categories, helps understand the severity and immune response of affected patients. On the other hand, the operational classification simplifies the identification of cases for treatment and disease control purposes.

According to Santarém (2018), the Master Plan, established by Law n˚. 20,534 defines the territorial division of the city into 5 zones in the urban zone, areas of urban expansion - corresponding to the territory available for urbanization - and 5 neighborhoods in the rural zone, in Table 3, the Ranking of Leprosy Cases by Unit was highlighted Notifier of the Urban Zone of the City of Santarém (2011–2020), with the highest number of cases in the central zone. The city of Santarém has 48 neighborhoods in the following zones: North (Caranazal, Liberdade, Mapiri, Laguinho, Fátima, Aparecida, Centro, Santa Clara, Aldeia, Santíssimo and Prainha); Central (Esperança, Aeroporto Velho, Jardim Santarém, Interventoria, Diamantino and Floresta); East (Livramento, Uruará, Área Verde, Jutaí, Urumari, Maicá, Pérola do Maicá, Jardelândia, Urumanduba, Santana and São José Operário); West (Amparo, São Cristóvão, Alvorada, Conquista, Novo Horizonte, Santarenzinho, Maracanã, Maracanã I, Nova Jerusalem, Nova Vitória and Elcione Barbalho); and South (Cambuquira, Ipanema, Mararu, Vigia, Vitória Régia, Nova República, Matinha, São Francisco and Santo André).

The analysis of Relative Risk (RR) concerning neighborhoods in the urban area of the City of Santarém is presented in Fig 1. It is noteworthy that the Floresta neighborhood has a Relative Risk of 3.11, Confidence Interval (1.87, 5.18); Fátima neighborhood: RR = 2.31,

**Table 2. Leprosy Description by Clinical and Therapeutic Variables According to Year of Notification in Santarém City (2011–2020).**

| VARIABLES CLINICS | 2011 | | 2012 | | 2013 | | 2014 | | 2015 | | 2016 | | 2017 | | 2018 | | 2019 | | 2020 | | TOTAL | |
|---|---|---|---|---|---|---|---|---|---|---|---|---|---|---|---|---|---|---|---|---|---|---|
| | N | % | n | % | n | % | No | % | n | % | n | % | n | % | n | % | n | % | n | % | n | % |
| **Classification Operational** | | | | | | | | | | | | | | | | | | | | | | |
| Paucibacillary | 24 | 34 | 18 | 29 | 9 | 16 | 19 | 23 | 13 | 20 | 22 | 33 | 13 | 25 | 9 | 17 | 13 | 25 | 10 | 42 | 150 | 26 |
| Multibacillary | 47 | 66 | 44 | 71 | 47 | 84 | 64 | 77 | 52 | 80 | 44 | 67 | 38 | 75 | 43 | 83 | 38 | 75 | 14 | 58 | 431 | 74 |
| *p-value* | 0.009 | | 0.0015 | | <0.001 | | <0.001 | | <0.001 | | 0.009 | | 0.0008 | | <0.001 | | 0.0008 | | - | | - | |
| **Clinical form** | | | | | | | | | | | | | | | | | | | | | | |
| Not filled | 1 | 1 | 0 | 0 | 2 | 4 | 1 | 1 | 0 | 0 | 0 | 0 | 0 | 0 | 1 | 2 | 1 | 2 | 4 | 17 | 10 | 2 |
| Indeterminate (IL) | 9 | 13 | 7 | 11 | 9 | 16 | 11 | 13 | 4 | 6 | 9 | 13 | 6 | 13 | 4 | 8 | 8 | 16 | 5 | 21 | 72 | 13 |
| Tuberculoid (TL) | 15 | 21 | 11 | 18 | 3 | 5 | 9 | 11 | 10 | 16 | 10 | 15 | 7 | 16 | 4 | 8 | 5 | 10 | 2 | 8 | 76 | 13 |
| Borderline (DL) | 31 | 44 | 31 | 50 | 30 | 53 | 41 | 50 | 32 | 50 | 31 | 46 | 19 | 42 | 24 | 46 | 20 | 39 | 6 | 25 | 265 | 46 |
| Lepromatous (LL) | 14 | 20 | 13 | 21 | 11 | 19 | 18 | 22 | 13 | 20 | 13 | 19 | 13 | 29 | 18 | 35 | 17 | 33 | 7 | 29 | 137 | 24 |
| Not classified | 1 | 1 | 0 | 0 | 2 | 4 | 2 | 2 | 5 | 8 | 4 | 6 | 0 | 0 | 1 | 2 | 0 | 0 | 0 | 0 | 15 | 3 |
| *p-value* | <0.001 | | <0.001 | | <0.001 | | <0.001 | | <0.001 | | <0.001 | | <0.001 | | <0.001 | | <0.001 | | - | | - | |
| **Number of skin lesions** | | | | | | | | | | | | | | | | | | | | | | |
| Not Informed | 3 | 4 | 0 | 0 | 12 | 21 | 7 | 9 | 12 | 18 | 9 | 14 | 11 | 22 | 10 | 19 | 6 | 12 | 5 | 21 | 75 | 13 |
| One lesion only | 15 | 21 | 15 | 24 | 9 | 16 | 30 | 37 | 20 | 31 | 17 | 26 | 7 | 14 | 14 | 27 | 10 | 20 | 5 | 10 | 142 | 24 |
| 2–5 lesions | 27 | 38 | 18 | 29 | 13 | 23 | 32 | 39 | 17 | 26 | 15 | 23 | 13 | 25 | 10 | 19 | 12 | 24 | 9 | 17 | 166 | 29 |
| > 5 lesions | 26 | 37 | 29 | 47 | 23 | 40 | 13 | 16 | 16 | 25 | 25 | 38 | 20 | 39 | 18 | 35 | 23 | 45 | 5 | 10 | 198 | 34 |
| *p-value* | <0.001 | | <0.001 | | 0.051 | | <0.001 | | 0.569 | | 0.047 | | 0.073 | | 0.336 | | 0.006 | | - | | - | |
| **No. of Nerves affected** | | | | | | | | | | | | | | | | | | | | | | |
| 0 nerves | 47 | 66 | 42 | 68 | 44 | 79 | 58 | 70 | 48 | 74 | 45 | 68 | 27 | 53 | 38 | 73 | 29 | 57 | 18 | 75 | 396 | 68 |
| 1 to 3 nerves | 18 | 25 | 15 | 24 | 11 | 20 | 23 | 28 | 11 | 17 | 18 | 27 | 21 | 41 | 10 | 19 | 16 | 31 | 5 | 21 | 148 | 25 |
| 4 to 7 nerves | 6 | 8 | 5 | 8 | 1 | 2 | 2 | 2 | 6 | 9 | 3 | 5 | 3 | 6 | 4 | 8 | 6 | 12 | 1 | 4 | 37 | 6 |
| *p-value* | <0.001 | | <0.001 | | <0.001 | | <0.001 | | <0.001 | | <0.001 | | 0.0001 | | <0.001 | | 0.0004 | | - | | - | |
| **TOTAL** | **71** | **100** | **62** | **100** | **56** | **100** | **83** | **100** | **65** | **100** | **66** | **100** | **51** | **100** | **52** | **100** | **51** | **100** | **24** | **100** | **581** | **100** |

**Source:** Secretary City of Health in santarem (SEMSA), Santarém/PA, 2021.

Confidence Interval (1.36, 3.95); Nova República neighborhood: RR = 3.55, Confidence Interval (2.09, 6.05); Matinha neighborhood: RR = 2.41, Confidence Interval (1.42, 4.10); Santarenzinho neighborhood: RR = 2.15, Confidence Interval (1.26, 3.66), reinforcing the significance of these associations. These results suggest that these neighborhoods require special attention in the development of preventive and control strategies for Hansen's disease.

## Discussions

This finding aligns with a study by Da Paz et al. [31], which reported an 11,357 reduction in leprosy cases in Brazil during 2020, representing a 41% decrease compared to the average number of cases between 2015 and 2019.

According to Jardim et al. [32], the pandemic brought about changes in the care profile for patients with chronic diseases across different levels of healthcare. This led to the cancellation and subsequent rescheduling of appointments, resulting in challenges for diagnosis and follow-up for patients, including those with leprosy. Consequently, this process facilitated the worsening of their health conditions, development of reactions, and disabilities.

It was observed that 60% of the patients were male, consistent with the predominance of affected men reported by Brasil [33] in the Epidemiological Bulletin on Leprosy. Additionally,

**Table 3. Ranking of Leprosy Cases per Notifying Unit in the Urban Zone of Santarém City (2011–2020).**

| ZONE URBAN | | | |
|---|---|---|---|
| **ZONE CENTRAL** | | **ZONE NORTH** | |
| UBS da Floresta | 45 | UBS Fatima | 22 |
| UBS Esperança | 22 | UBS Aparecida-Caranazal | 20 |
| UBS Diamantino | 20 | UBS Mapiri-Liberdade | 18 |
| UBS Aeroporto Velho | 14 | UBS Santa Clara | 11 |
| UBS Jardim Santarém | 11 | UBS Santíssimo Prainha | 5 |
| UBS Interventoria | 8 | **TOTAL** | **76** |
| URE Santarém | 5 | | |
| Hospital Municipal de Santarém | 4 | **ZONE WEST** | |
| Centro de Saúde Universitária | 2 | UBS Santarenzinho | 37 |
| HRBA | 2 | UBS Maracanã | 31 |
| | | UBS Conquista | 15 |
| **TOTAL** | **133** | **TOTAL** | **83** |
| **ZONE SOUTH** | | **ZONE EAST** | |
| UBS Nova República | 48 | UBS Santana | 20 |
| UBS Matinha | 23 | UBS Maicá | 19 |
| UBS Santo André | 8 | UBS Jaderlândia | 18 |
| UBS Vitoria Regia | 8 | UBS Uruará | 16 |
| UBS Mararu | 3 | UBS Área Verde | 9 |
| | | UBS Livramento | 8 |
| | | UBS Jutaí Urumarí | 3 |
| **TOTAL** | **90** | **TOTAL** | **93** |

**Source:** Secretary of Health in Santarem City (SEMSA), Santarém/PA, 2021.

**Caption**: UBS: Unit Basic in Health; HRBA: Hospital Regional of Low Amazon; URE: Unit in Reference Specialized

men often delay seeking healthcare, hindering the early interruption of the disease's clinical progression, resulting in late diagnosis and irreversible physical deformities [34].

Analyzing the distribution of cases by age group, it was observed that the majority of patients (94%) were aged 15 or over, with only 6% under 14 years of age. According to Brito Moreira et al. [35], leprosy is considered a disease affecting adults and young adults, and due to its evolution with a high degree of physical disability, it can cause significant social harm to patients, especially those in economically active age groups, while also affecting their mental health due to the associated social stigma.

The significant number of Hansen's disease cases in patients under 14 years of age raises concerns about the potential expansion of the bacillus transmission chain within the household environment of these individuals, which may indicate shortcomings in health surveillance, as pointed out by Souza et al. [36]. The occurrence of the disease in children is often linked to recent infections and the presence of multiple active transmission foci within family clusters and the community. This underscores the importance of effective measures for prevention, early detection, and treatment of Hansen's disease, particularly in areas where the disease remains endemic, as is the case in the city of Santarém. These concerns are supported by the study conducted by Barreto et al. [37] on Hansen's disease in children, which addressed the challenges faced in controlling the disease in this age group.

In the new technical guidelines from the World Health Organization (WHO) on the interruption of transmission and elimination of leprosy in 2023, a significant shift is

observed in the criteria for defining the interruption of transmission. Notably, the criterion of "zero pediatric cases for five consecutive years" has emerged as the key indicator. It highlights the collective efforts in disease control and signifies a remarkable milestone on the path to leprosy eradication [38,39].

In terms of the racial variable, the majority of leprosy cases involved individuals of brown ethnicity, accounting for 74% of the cases. The observed variation in the distribution of skin colors hints at a complex tapestry of migration and demographics in Brazil. These findings align with several other studies, including those conducted by Anjos et al. [34], and Oliveira [40], wherein a predominant proportion of leprosy patients were of mixed-race backgrounds. This phenomenon can be attributed to Brazil's ethnically diverse composition resulting from extensive intermixing of ethnicities over time. Additionally, the higher incidence of leprosy among individuals with darker skin tones may signal underlying socioeconomic disparities that render these populations more susceptible to the disease, owing to substandard living conditions, limited access to healthcare, and a lack of health education.

Hence, it is imperative to address leprosy not merely as a public health challenge but also as a reflection of the enduring social and economic inequalities in Brazil. The implementation of comprehensive healthcare policies and the expansion of medical services accessibility in remote regions, such as the Amazon, represent pivotal measures to curtail the disease's cumulative incidence and enhance the well-being of traditional populations [41,42]. Moreover, the development of educational initiatives aimed at bolstering awareness and early leprosy diagnosis, particularly within vulnerable communities, should be an integral component of a broader endeavor to combat healthcare disparities in the nation. Additionally, it is worth noting that in the North region, leprosy also affects the indigenous population, constituting 1% of the cases. Teles et al. [43] have highlighted the possibility of underreporting within this demographic due to their limited access to healthcare services.

As of the last IBGE census in 2010, the city of Santarém, located in the Lower Amazon Mesoregion, covered an area of 22,890 $km^2$ and had a population of 294,580 inhabitants. Currently, it is estimated to have 308,339 inhabitants, according to IBGE [44]. Analyzing the reported cases according to the patients' residential zones, the majority of cases occurred in the urban zone, followed by the rural zone, which is in agreement with the study by Da Silva Vieira et al. [45], which also showed a cumulative incidence of cases in the urban zone. According to Da Silva et al. [46], the majority of cases concentrate in the urban zone due to the larger population size, but it is important to note the possibility of underreporting in rural areas and the large number of medical records with this field left blank, indicating the need for improvements in healthcare services.

The combination of Ridley-Jopling classifications and operational classification allows for a more comprehensive analysis of leprosy distribution over the years, providing valuable insights into the predominant clinical forms, bacterial burden, number of skin lesions, and nerve involvement. This is crucial for guiding public health policies and leprosy control strategies in an endemic area like Santarém [47,48]. Furthermore, while the operational classification is primarily used for treatment follow-up in Brazil, it is essential to analyze Ridley-Jopling classification to gain a deeper understanding of the epidemiological landscape. Ridley-Jopling categories provide insights into the severity of the disease and the immune response of affected individuals, contributing to a more nuanced assessment of leprosy cases.

By comparing these classifications over the years, we can identify trends and changes in the disease's epidemiology, allowing for targeted efforts in areas of greater need. Therefore, the inclusion of both classifications enhances the quality of epidemiological analysis and strengthens the effectiveness of public health interventions.

Regarding the operational classification and number of skin lesions, 72% of leprosy patients in Santarém had MB leprosy, which was the most prevalent form throughout the analyzed period, and 34% presented more than five dermatological lesions. However, at least 53% of the sample presented fewer than five lesions at the time of diagnosis. This data is in line with Brasil [33], which reported 80.1% new MB cases in the country, and Pará reported 80%, indicating an increase in MB detection over the years.

The study by Da Cruz et al. [49] supports these findings, as they observed a higher number of patients affected by the Dimorphous Leprosy type (72.08%) and Lepromatous Leprosy type (22.31%). Pereira et al. [50] also present similar data, where 69.91% were infected with the Dimorphous Leprosy form, followed by the Lepromatous Leprosy form (18.29%), as well as the data presented by Melo et al. [51], who observed higher frequencies in the Dimorphous Leprosy and Tuberculoid Leprosy forms, totaling an average of 44.9% of cases. However, the study by Lima et al. [52] showed a discrepancy with the present study, as their data had Lepromatous Leprosy as the predominant form (52.2%) in relation to Dimorphous Leprosy with 18.8%, but it agrees with the lower frequency of the Indeterminate Leprosy and Tuberculoid Leprosy clinical forms.

This pathology is negatively impacted by the delay in diagnosing patients with leprosy, caused by several factors. According to Melão [53], this chronic and advanced scenario of the disease reflects late diagnosis and difficulty of access. De Souza et al. [54] state that such patients are in more vulnerable situations, have a higher possibility of developing disabilities, and are also carriers of contagious forms of the infection, acting as transmitters before starting treatment. This notion is in line with Pereira et al. [55], as concerning the clinical form, the low frequency of IL, and the high rate of more severe forms can also be explained by delayed diagnosis, lack of information in society, and the patient's difficulty in accepting that they have leprosy, allowing the disease to progress to more severe forms until the patient finally seeks medical assistance.

Regarding the number of affected nerves at the time of diagnosis, the majority of patients, 396 (68%) of the cases, had no affected nerves, and only 37 (6%) had more than 4 affected nerves. This result is consistent with De Santana et al. [56], who found 52.4% of patients without affected nerves. It was observed that 31% of patients already have nerve damage. This data serves as a warning to healthcare services, as leprosy is a debilitating disease with a high degree of physical disability and a high likelihood of causing sequelae, which can interfere with the social life of its carriers and lead to various types of psychological trauma [57].

According to Vieira [58], primary care serves as the entry point to the healthcare system, coordinating actions and services for the population. As such, Primary Health Care (PHC) practices play a vital role in actively searching for patients with infectious diseases like leprosy. The strengthening of PHC in the Leprosy Control and Elimination Program occurred primarily after the publication of the Operational Norm of Health Care (NOAS) 01/2001 by the SUS (Unified Health System), which decentralized health care and prioritized leprosy as a focus disease. UBSs (Basic Health Units) equipped with better resources and well-organized teams for active case finding tend to have higher case detection rates, as noted by Pinho [59].

The study's findings emphasize the critical nature of disability and the ramifications of delayed leprosy diagnosis, reinforcing the necessity for proactive early detection and educational programs. These measures are essential to lessen the long-term effects of leprosy, highlighting the role of preventative and educational strategies in addressing the disease's consequences. Our research aligns with findings by Santos et al. [60], which identify factors contributing to leprosy's delayed diagnosis in Northeastern Brazil, stressing the importance of these interventions in endemic regions. The study underscores that early detection and

informed public health strategies can significantly reduce leprosy's burden, demonstrating a clear path forward in mitigating its long-standing impacts.

Analyzing the spatial distribution of leprosy cases, Da Silva Baia [61] highlights that the disease does not occur randomly; the highest incidence is concentrated in more deprived areas. This type of analysis helps reveal patterns in the most affected regions and correlates them with socio-environmental factors. It also contributes to improving family health strategies, promoting training, and enhancing coverage to achieve greater prevention and health promotion in Santarém.

The analysis of Relative Risk (RR) for neighborhoods in the urban area of the City of Santarém, it has been noted that the spatial clustering tool depicted neighborhoods Perola and Maica as being in a low-risk cluster, despite these neighborhoods individually exhibiting a higher relative risk. This observation is an excellent opportunity to discuss the inherent methodological nuances involved in spatial risk assessments. Localized risk assessments are sensitive to specific neighborhood conditions and are influenced by small-scale, local variables. This granularity allows us to detect acute risk factors that may be unique to a particular area. For example, neighborhoods like Pérola and Maica have specific environmental and social determinants that increase their individual risk levels..

Conversely, spatial clustering operates on a more macro scale. By aggregating data from contiguous neighborhoods, this method aims to discern patterns that transcend individual neighborhood boundaries. The tool is particularly useful for identifying larger areas of uniform risk, which can be crucial for resource allocation and strategic planning. However, this aggregation can sometimes obscure the localized nuances, presenting a smoothed-over view that might not reflect small-scale heterogeneities. This difference in scale and the resulting interpretation underscores the complexity of spatial risk analysis. High-risk neighborhoods can be situated within low-risk clusters due to the averaging effect of spatial clustering, and vice versa.

Notably, the Floresta and Nova República neighborhoods, situated at a greater distance from the urban center, exhibit a substantial RR of 3.11 and 3.55, respectively, indicating a heightened risk of disease incidence. The Confidence Intervals further underscore the statistical significance of these associations. The geographical remoteness of these neighborhoods poses challenges to accessing healthcare services, potentially contributing to the elevated risk observed. These findings emphasize the importance of considering spatial factors and healthcare accessibility in the development of targeted preventive and control strategies for Hansen's disease, particularly in more distant neighborhoods where additional efforts may be required to ensure equitable health outcomes [62].

In light of this, we suggest that future analyses could benefit from a dual-scale approach. By assessing risk at both the local (neighborhood) and regional (cluster) levels, we can provide a more nuanced and comprehensive view of risk distribution. This multi-scale analysis would allow us to appreciate the micro-variations within broader trends, providing a richer and more informative landscape for decision-makers.

## Conclusion

The characterization of leprosy in Santarém from 2011 to 2020 revealed that the majority of patients were male, aged 15 years or older, of mixed race, and residing in urban areas. In terms of clinical variables, there was a cumulative incidence of patients classified as multibacillary, more than 5 skin lesions, and without affected nerves. The difficulty of accessing health services in these more remote areas may contribute to a delayed detection of the disease. As a

study limitation, there was no systematic contact tracing conducted, which could indicate a potentially unmet demand.

The research also highlighted the profound and insidious impact of the COVID-19 pandemic on leprosy control in the city. Thus, prioritizing patient care and reducing barriers to accessing health services should be a priority for governments and health managers. The study's originality and significance stem from its capacity to offer a multidimensional perspective on leprosy, thereby assisting policymakers and healthcare professionals in making informed decisions to effectively combat this endemic disease. This research not only contributes to our understanding of leprosy dynamics but also underscores the necessity of adaptive strategies in response to the challenges posed by concurrent health crises.

## Author Contributions

**Conceptualization:** Edson Jandrey Cota Queiroz, Ingrid Nunes da Rocha, Lívia de Aguiar Valentim, Caroline Gomes Macedo, Tatiane Costa Quaresma.

**Formal analysis:** Edson Jandrey Cota Queiroz, Ingrid Nunes da Rocha, Lívia de Aguiar Valentim, Caroline Gomes Macedo, Tatiane Costa Quaresma.

**Investigation:** Edson Jandrey Cota Queiroz, Ingrid Nunes da Rocha, Lívia de Aguiar Valentim, Caroline Gomes Macedo, Tatiane Costa Quaresma.

**Methodology:** Edson Jandrey Cota Queiroz, Ingrid Nunes da Rocha, Lívia de Aguiar Valentim, Caroline Gomes Macedo, Tatiane Costa Quaresma.

**Writing – original draft:** Edson Jandrey Cota Queiroz, Ingrid Nunes da Rocha, Lívia de Aguiar Valentim, Caroline Gomes Macedo, Tatiane Costa Quaresma.

**Writing – review & editing:** Edson Jandrey Cota Queiroz, Ingrid Nunes da Rocha, Lívia de Aguiar Valentim, Thiago Junio Costa Quaresma, Zilmar Augusto de Souza Filho, Sheyla Mara Silva de Oliveira, Franciane de Paula Fernandes, Caroline Gomes Macedo, Tatiane Costa Quaresma, Waldiney Pires Moraes.

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
