## [Decision Letter · Decision Letter 0]

28 Aug 2023

Dear Dr. Valentim,

Thank you very much for submitting your manuscript "Epidemiological, Clinical, and Geographical Characterization of Leprosy in the Municipality of Santarém-Pará: Insights for Effective Control and Targeted Intervention" for consideration at PLOS Neglected Tropical Diseases. As with all papers reviewed by the journal, your manuscript was reviewed by members of the editorial board and by several independent reviewers. In light of the reviews (below this email), we would like to invite the resubmission of a significantly-revised version that takes into account the reviewers' comments. 

Your manuscript has been reviewed by three experts in the field. There are major concerns expressed by all three reviewers. Should you decide to re-submit the manuscript, there will be major changes and improvements required to make the manuscript acceptable for publication in PLoS Neglected Tropical Diseases. Please address each issue fully and provide the necessary details.

We cannot make any decision about publication until we have seen the revised manuscript and your response to the reviewers' comments. Your revised manuscript is also likely to be sent to reviewers for further evaluation.

Sincerely,

Paul J. Converse

Academic Editor

Ana LTO Nascimento

Section Editor

Dear Dr. Livia Valentim,

Your manuscript has been reviewed by three experts in the field. There are major concerns expressed by all three reviewers. Should you decide to re-submit the manuscript, there will be major changes and improvements required to make the manuscript acceptable for publication in PLoS Neglected Tropical Diseases. Please address each issue fully and provide the necessary details.

Reviewer's Responses to Questions

**Key Review Criteria Required for Acceptance?**

**Methods**

-Are the objectives of the study clearly articulated with a clear testable hypothesis stated?

-Is the study design appropriate to address the stated objectives?

-Is the population clearly described and appropriate for the hypothesis being tested?

-Is the sample size sufficient to ensure adequate power to address the hypothesis being tested?

-Were correct statistical analysis used to support conclusions?

-Are there concerns about ethical or regulatory requirements being met?

Reviewer #1: This paper is simply a retrospective description of leprosy cases in one area of Brazil over a 10 year period. There are no original insights presented.

On line 84, it is stated that Brazil had 45 cases per 100,000 pop in 2019. According to my calculations based on the WHO data, Brazil had 31,827 prevalent cases and 27,863 new cases in 2019, in a total population of 210m. This gives a prevalence rate of 15 per 100,000 and a new case detection rate of 13 per 100,000.

Reviewer #2: The objectives of the study are clearly articulated, namely to provide an epidemiological, clinical and geographic characterization of leprosy in the northern Brazilian city of Santarem during the period 2011-2020.

The study design is appropriate to address the stated objective.

The population is clearly described and appropriate with the sample size large enough.

Statistical analyses seem appropriate to support conclusions.

No ethical or regulatory concerns.

Reviewer #3: The methods described do not match what is presented in the results very well.

**Results**

-Does the analysis presented match the analysis plan?

-Are the results clearly and completely presented?

-Are the figures (Tables, Images) of sufficient quality for clarity?

Reviewer #1: There was no analysis plan, other than to describe the cases on record.

Racial and geographical variations in the leprosy incidence are mentioned and could have provided some interesting discussion points, but were not analysed in any detail.

Reviewer #2: The analysis presented matches the analysis plan. 

Some changes need to be made regarding the forms of leprosy as there appears to be a mixture of Ridley-Jopling and forms of leprosy peculiar to Brazil that might be confusing to readers outside of Brazil. 

The names of some of the Basic Health Units that are translated into English should be returned to their original names in Portuguese to make them easier to find.

Reviewer #3: Yes, it does

No, the results are presented clearly. It is very confusing that results and discussion are mixed together.

No, the tables would need considerable work

Line 150: it is not clear why Results and Discussion are mixed. The normal format is to present results first separately and then interpret them and put them in the context of recent literature in the Discussion. This is now blurred together.

**Conclusions**

-Are the conclusions supported by the data presented?

-Are the limitations of analysis clearly described?

-Do the authors discuss how these data can be helpful to advance our understanding of the topic under study?

-Is public health relevance addressed?

Reviewer #1: There were no original discussion points or conclusions.

Reviewer #2: The conclusions are mainly supported by the data. 

The authors discuss how the distribution of the cases of leprosy in the 48 neighborhoods could be used to identify hot spots and be helpful to focus the efforts of health officials towards directing more efforts to these areas in leprosy control.

Reviewer #3: No clear conclusions are drawn, just a description of the distribution of leprosy cases in the study period.

Line 327-329: The spatial analysis seems to be missing; this would have been the most salient feature of this paper.

**Editorial and Data Presentation Modifications?**

Reviewer #1: (No Response)

Reviewer #2: Minor points

In Table 1, there were several errors. Instead of the heading “Race” replace with “Skin color”. Unclear what “Ign/White” means under Race and Zone in residence. There were several discrepancies in the values of the number of cases and percent, in Ign/White for 2019, 0 = number does not equal 4%; in the total in this line 4%5 probably should be n = 4; in the Yellow line, 1 = number should be 2% but it also is found 1 = number and 1% in 2014 and 0 = number does not equal 2% in 2019. The total percent in 2019 adds up to 142%, it should be 100%.

Line 179 “with the highest number observed in 2014 (74%) and the lowest in 2020 (83%)” replace percentage with number of cases, for 2014 n = 74 and for 2020 n = 20

Lines 184-186 “The significant number of patients under 14 years old is worth noting, which could be attributed to an increase in the transmission chain of the bacillus in the household environment of these individuals, indicating a deficiency in health surveillance”. Cases in children are usually attributed to recent disease and multiple active foci of transmission in households and the community, see Barreto JG et al. (2017) Leprosy in children. Curr Infect Dis Rep 19:23.

Table 2 under “Form Clinic” (Clinical form?), clarify what ign/white means. “undertemined” should probably be “Indeterminate”. “No classified” should be “Not classified”

Table 2 under “No. in injuries” should probably be corrected to “Number of skin lesions” Unclear what the meaning is “Informed 0 or 99” (no lesions?); “Lesion only” maybe should be “one lesion only”; “2-5 injuries” and “>5 injuries” should be changed to “lesions”

Lines 227-228, unclear what the meaning of “HD type” and “HD form”. Line 232 and 238, unclear what “HI” means (Indeterminate?)

Lines 243-244 “only 37 (6%) presented lesions in more than 4 nerves” should be “only 37 (6%) had more than 4 affected nerves”

In lines 251-261, the Brazilian names for neighborhoods in the various zones and the names for the Basic Health Units (UBS) shown in the map of Santarem in Figure 1 appear to be translated into English in Table 3 which is a little confusing since not all of the names translate well. It would be better to just leave the names in Table 3 as the original Brazilian names found in the map, makes all of them easier to find.

Reviewer #3: Abstract

Line 32: I’m not familiar with Santarém-Pará, but quite a few other municipalities I have visited in Brazil are not only a city but include some rural areas also. Given the situation of Santarém-Pará, would ‘municipality’ be a better term or ‘city’ (which suggests only an urban area)?

Line 36: ‘prevalence’. The denominator is the cumulative incidence of new cases detected in the study period? If so, it would worth mentioning this, since prevalence usually has a population denominator and is annual, rather than cumulative over a long period. 

Line 38: Assuming that no ‘old cases’ from before the study period were included, this ‘prevalence’ should be cumulative incidence. This should be corrected elsewhere in the manuscript also.

Introduction

Line 83/84: “Currently, leprosy is considered endemic in countries where the prevalence exceeds ten cases 84 per 100,000 inhabitants”. With the release of the WHO Technical guidance on interruption of transmission and elimination of leprosy (2023), this definition/cut-off is now no longer current. 

Results

Line 150: it is not clear why Results and Discussion are mixed. The normal format is to present results first separately and then interpret them and put them in the context of recent literature in the Discussion. This is now blurred together.

Table 1

It is not clear what the p-values relate to – comparing percentages? If so, it not clear what the purpose of that would be. Also, there are way too many p-values in this table. This would require a Bonferroni correction. But better would be to omit the statistical testing and present the percentages on a separate row, so the trends become visible more easily.

Line 171: Ref to Basso – this is at best an hypothesis, since this has never been be demonstrated in immunological studies. In several other population based studies, when active case detection was employed, the proportion of female and male cases was about equal.

Line 172 and following: this is again one possible explanation. It is also possible that that men do jobs that cause injuries to their limbs more easily.

Line 176: if one age group is >15, the other should be 15 or below, not below 14. But since the normal cut-off used for children by WHO and also in SINAN is <15, should this not be cut-off? So <15 and 15 and above for adults?

“These results are similar to those of Da Silva Vieira et al. (2020), who also reported a prevalence of leprosy in individuals above 15 years old.” The meaning of this sentence is not clear. Most leprosy cases are reported in adults in any study or epidemiological report, so what is the result here that is pertinent to this study? The trend in child cases would be more interesting to report, since ‘zero autochthonous child cases for 5 consecutive years’ is now the indicator defining interruption of transmission in the new WHO technical guidance on interruption of transmission and elimination of leprosy (2023). 

Line 185: why an increase in transmission? What is the evidence of that in the data presented? Or do the authors mean ‘a relatively high level of transmission in the household.’?

Line 187/88: When reporting small numbers of cases, like the child cases in this study, it is preferable to look at absolute numbers rather than percentages. Also, the year should be 2014, not 2009.

Line 201: ‘prevalence’ should be ‘predominance’? Prevalence just means occurrence, which would be meaningless in this sentence. Is the difference cumulative incidence in these different areas not just a reflection of the population figures, presence of health care facilities or other access factors that make it more difficult for rural populations to access leprosy services? The latter is already noted by the authors.

Lines 205-2012: these are neither results or discussion of results.

Table 2: as with the above, the p-value can omitted and the percentages are best place on a separate row.

Meaning of ‘ign/white’ is not clear

Meaning of ‘Informed 0-99’ is not clear

To what extent are all the studies that are cited population-based studies? If they concern a single centre, rather than an entire municipality, bias is likely and comparisons with the current study cannot be made.

Lines 240-241: it is definitely true that a proportion of patients will develop more extensive disease if diagnosis and treatment are delayed. However, this is not the only reason for a high MB proportion. Milder forms of leprosy such as PB and mild BT are likely to self-heal, if not diagnosed in time. This will result in an apparent high MB proportion. Also, when transmission has stopped or is declining, the proportion of MB will increase, since MB disease has a longer incubation period than PB.

Line 263-261: these are not interesting for readers outside the municipality. This could either be omitted or moved to an annex. Same for Table 3.

Figure 1 could be omitted or moved to an annex

What would be interesting is to see the cases GIS mapped and then submitted to a cluster analysis.

Conclusion

No clear conclusions are drawn, just a description of the distribution of leprosy cases in the study period.

Line 327-329: The spatial analysis seems to be missing; this would have been the most salient feature of this paper.

**Summary and General Comments**

Reviewer #1: This is a very weak paper, without any original ideas.

Reviewer #2: This study provides an epidemiological, clinical and geographic characterization of leprosy in the city of Santarem, PA during the period from 2011-2020. There were 581 cases of leprosy diagnosed with the following distribution: male (60%), age over 15 years (94%), brown skin (74%), urban area (73%), multibacillary (74%), no nerves affected (68%). Higher numbers were found in the city’s central area with 133 cases and the highest number of cases (48) reported coming from the basic health unit of Nova Republica in the south. These features can be used to see if there is a relationship between socio-economic conditions and incidence rates in particular neighborhoods of Santarem. The following comments and questions should be addressed. 

1. In the characterization of race for Brazilians, it should be clarified that the mention of “brown race” in Table 1 or “mixed race” in the Conclusion is a bit misleading since individuals with brown skin, very common in the northern state of Para, cannot be used to identify complex racial mixtures in these people. This has been well-studied in the Brazilian population that has a high degree of admixture from White (European), Black (African) and Amerindian that makes self-declarations of ethnicity impossible to confirm without identifying mitochondrial haplogroups (mtDNA) and genomic ancestry (Cardena MMSG et al. 2013. PLoS ONE 8: e62005). In this study, around 25% of those who self-reported as “pardo” or “brown” had significant Amerindian mtDNA with a similar contribution of European and African genomic ancestries. Please change “Race” to “Self-reported skin color”. Certainly, the overall high percentage of “brown skin” at 74% likely reflects somewhat higher admixtures of Black and Amerindian genomic ancestry that is common in Para state.

2. Lines 205-212, the description of the forms of leprosy are a mixture of Ridley-Jopling with polar tuberculoid (TT) and Dimorphous Leprosy (DL), a form description that is commonly used mainly in Brazil and usually includes the unstable borderline forms (BT, BB and BL) and Virchowian Leprosy (VL), also commonly used mainly in Brazil, usually denoting polar lepromatous leprosy (LL). Mixing and matching these terms might be confusing for readers outside of Brazil more familiar with the five Ridley-Jopling forms (TT, BT, BB, BL and LL). 

3. It would be good to show the new case detection rate (NCDR) for Santarem during this time period that likely would be considered highly endemic or hyperendemic. The NCDR in the state of Para has been known to be hyperendemic in many cities (>4.0 per 10,000 population) and is only exceeded in states like Mato Grosso and Maranhao.

4. A number of groups have published the location of leprosy patients and other groups (affected schoolchildren, household contacts) in various cities in Brazil using a Geographic Information System and spatial epidemiological tools to map hotspots of leprosy occurring in neighborhoods within a city over time (see Nicchio MVC et al. 2016. Acta Trop 163: 38-45; Queiroz JW et al. 2010. Am J Trop Med Hyg 82: 306-314; Barreto JG et al. 2014. PLoS Negl Trop Dis 8: e2665; Barreto JG et al. 2015. BMC Inf Dis 15: 527). It would have been much better to map the GIS location of all of the 581 individuals on the map in Figure 1 that would more clearly show the distribution of the cases found in the 48 individual neighborhoods listed in lines 251-261. This would really show where the hotspots of leprosy are occurring in specific neighborhoods of Santarem allowing for more targeted approaches to leprosy control.

Reviewer #3: The paper is not really a research paper. It is a helpful report describing the distribution of leprosy cases detected in one municipality in Pará State in northern Brazil. Despite the fact that spatial analyses were announced in the Methods, no results of these are presented. There are no clear other salient features. The authors wrote: “The epidemiological, clinical, and geographical characterization of leprosy in the municipality of Santarém-Pará between 2011 and 2020, as presented in this study, can provide valuable insights for formulating more effective strategies for prevention, control, and targeted intervention in this specific region.” While this is true in general, it does not become how the specific data presented could be used for this purpose.

PLOS authors have the option to publish the peer review history of their article (what does this mean?). If published, this will include your full peer review and any attached files.

Reviewer #1: No

Reviewer #2: Yes: John S. Spencer

Reviewer #3: No

Figure Files:

Data Requirements:

Please note that, as a condition of publication, PLOS' data policy requires that you make available all data used to draw the conclusions outlined in your manuscript. Data must be deposited in an appropriate repository, included within the body of the manuscript, or uploaded as supporting information. This includes all numerical values that were used to generate graphs, histograms etc.. For an example see here: http://www.plosbiology.org/article/info:doi%2F10.1371%2Fjournal.pbio.1001908#s5.
---

## [Editor Report · Decision Letter 1]

12 Dec 2023

Dear Livia de Aguiar Valentim,

Before we can send the revised manuscript out for review, please address the two issues below:

1. You seem to misunderstand what is needed in “Author Summary.” You provided what seems to be your resumé instead of a lay reader abstract. Please look at the instructions to authors and other articles published in the journal.

2. Your rebuttal letter is a bit vague and general. Please give point-by-point responses, particularly in the case of reviewers 2 and 3, and indicate where in the revised manuscript those points are addressed.

We cannot make any decision about publication until we have seen the revised manuscript and your response to the reviewers' comments. Your revised manuscript is also likely to be sent to reviewers for further evaluation.

Sincerely,

Paul J. Converse

Academic Editor

Ana LTO Nascimento

Section Editor

Dear Livia de Aguiar Valentim,

Before we can send the revised manuscript out for review, please address the two issues below:

1. You seem to misunderstand what is needed in “Author Summary.” You provided what seems to be your resumé instead of a lay reader abstract. Please look at the instructions to authors and other articles published in the journal.

2. Your rebuttal letter is a bit vague and general. Please give point-by-point responses, particularly in the case of reviewers 2 and 3, and indicate where in the revised manuscript those points are addressed.

Figure Files:

Data Requirements:

Please note that, as a condition of publication, PLOS' data policy requires that you make available all data used to draw the conclusions outlined in your manuscript. Data must be deposited in an appropriate repository, included within the body of the manuscript, or uploaded as supporting information. This includes all numerical values that were used to generate graphs, histograms etc.. For an example see here: http://www.plosbiology.org/article/info:doi%2F10.1371%2Fjournal.pbio.1001908#s5.
---

## [Decision Letter · Decision Letter 2]

6 Feb 2024

Dear Dr. Valentim,

Thank you very much for submitting your manuscript "Epidemiological, Clinical, and Geographical Characterization of Leprosy in the County of Santarém-Pará: Insights for Effective Control and Targeted Intervention" for consideration at PLOS Neglected Tropical Diseases. As with all papers reviewed by the journal, your manuscript was reviewed by members of the editorial board and by several independent reviewers. In light of the reviews (below this email), we would like to invite the resubmission of a significantly-revised version that takes into account the reviewers' comments. 

Dear Dr. Agular Valentim,

Your revised manuscript has been reviewed by three experts in the field. They note both minor and major issues that need to be revised. Please carefully respond to each issue noted and alter the manuscript accordingly in a file with changes marked as well as a clean copy. Please note that the revisions suggested by Reviewer 5 are found in a separate file.

We cannot make any decision about publication until we have seen the revised manuscript and your response to the reviewers' comments. Your revised manuscript is also likely to be sent to reviewers for further evaluation.

Sincerely,

Paul J. Converse

Academic Editor

Ana LTO Nascimento

Section Editor

Dear Dr. Agular Valentim,

Your revised manuscript has been reviewed by three experts in the field. They note both minor and major issues that need to be revised. Please carefully respond to each issue noted and alter the manuscript accordingly in a file with changes marked as well as a clean copy. Please note that the revisions suggested by Reviewer 5 are found in a separate file.

Reviewer's Responses to Questions

**Key Review Criteria Required for Acceptance?**

**Methods**

-Are the objectives of the study clearly articulated with a clear testable hypothesis stated?

-Is the study design appropriate to address the stated objectives?

-Is the population clearly described and appropriate for the hypothesis being tested?

-Is the sample size sufficient to ensure adequate power to address the hypothesis being tested?

-Were correct statistical analysis used to support conclusions?

-Are there concerns about ethical or regulatory requirements being met?

Reviewer #2: The objectives of the study were clearly articulated; study design appropriate; population of study clearly defined; no ethical or regulatory concerns.

Reviewer #4: It is well described and the proper number included. But important information like disability and delay of diagnosis were not included. Basicly except the method there was nothing new, even the method not.

Reviewer #5: The study design is appropriate to state the objectives

The population is appropriate for the hypothesis being tested

The sample size is sufficient to ensure adequate power

The statistical analysis is appropriate to support conclusions

The concerns about ethical requirements were met.

**Results**

-Does the analysis presented match the analysis plan?

-Are the results clearly and completely presented?

-Are the figures (Tables, Images) of sufficient quality for clarity?

Reviewer #2: Results were clearly presented; Figures and Tables were good quality.

Reviewer #4: yes, but the analyses plan was not compleet.

Reviewer #5: The analysis presented match the analysis plan

The results need some clarification

The figures and tables need some clarification

**Conclusions**

-Are the conclusions supported by the data presented?

-Are the limitations of analysis clearly described?

-Do the authors discuss how these data can be helpful to advance our understanding of the topic under study?

-Is public health relevance addressed?

Reviewer #2: Conclusions supported by the data. Hot spots of leprosy cases detected in neighborhoods of Santarem can assist in targeted case finding interventions in the future.

Reviewer #4: Conclusions are supported. Limitatation not well described. Depended on by others collected data.

Reviewer #5: The conclusions are supported by the data presented

The limitations of the study are described

The authors discussed how the data will help to advance our understanding of the topic

Public health relevance is addressed

**Editorial and Data Presentation Modifications?**

Reviewer #2: This study provides an epidemiological, clinical and geographic characterization of leprosy in the city of Santarem, PA during the period from 2011-2020. There were 581 cases of leprosy diagnosed with the following distribution: male (60%), age over 15 years (94%), brown skin (72%), urban area (73%), multibacillary (74%), no nerves affected (68%). Higher numbers were found in the city’s central area with 133 cases and the highest number of cases (48) reported coming from the basic health unit of Nova Republica in the south. These features can be used to see if there is a relationship between socio-economic conditions and incidence rates in particular neighborhoods of Santarem. The following comments and questions should be addressed. 

1. The description of the clinical forms in lines 183-184 and Table 2 have some undefined abbreviations that Brazilian leprosy clinicians may be more familiar with but these names and abbreviations may be unfamiliar or confusing to leprosy researchers outside of Brazil. For clarity, it is really important to define the abbreviations in the text: DL (dimorphous leprosy, or borderline forms, which includes the three immunologically unstable borderline Ridley-Jopling forms borderline tuberculoid [BT], borderline borderline [BB] and borderline lepromatous [BL]); VL (Virchowian leprosy, or polar lepromatous LL form); IL (Indeterminate leprosy, an early stage of leprosy); and TL (tuberculoid leprosy, likely polar TT form). By clearly defining these clinical forms in the text, people will better understand what forms are included with the Clinical forms shown in Table 2 for Tuberculoid, Borderline and Virchowian.

2. The Abstract mentions “brown race” at 74% (this was not changed from the previous version) whereas in Table 1 it more correctly categorizes “Skin color – brown” at 72%.

3. The start of the paragraph “Discussion” begins abruptly with “This finding aligns…” unclear what this finding is in reference to, maybe this first sentence should be inserted elsewhere to provide better context. The next paragraph discusses how the pandemic resulted in a huge decrease in the new case detection rate, something that occurred worldwide, so maybe it should be placed here.

4. It is interesting that in Figure 2, two neighborhoods that appear on the map with the second highest risk (yellow), Perola and Maica, are shown as low risk on the spatial cluster map whereas Fatima in the north was calculated as having a high risk. Looks like the spatial clustering tool washed out the relatively high risk of these two neighborhoods. 

Minor point

Line 282, reference De Anjos et al. (2021) not found (Anjos et al.?)

Reviewer #4: Dear Authors,

I read your contribution and noticed I can only judge it as a clinician. The way you have shown is for me a black box. But I think it is an achievement that you showed what was well known.

Some remarks:

63: M. leprae and or M. lepromatosis 

64: in the skin also often Schwann cells particular Remak Schwann cells.

70: To be sure that it is leprosy you need 2 out of the 3 cardinal signs.

76: You can only speak of stopping the bacterial reproduction ( cure may be only at the paucibacillary site) At the lepromatous site you have often a relapse (often not seen because the follow-up is to short) or reinfection.

84: There is still a lot of discussion about the 3 drugs for PB and only ! year for MB particularly with a high BI. Or polar LL.

92: Not only of general public but of health professionals too.

104: I noticed no PREP. I fully agree. May be a place here for you to explain the reader why?

112; I see here that you also will focus on healthcare workers. But it is also essential to have a direct supporting team by WhatsApp or internet where they can get support.

139: Do not use race but background African, Indian, Asian, chinees, from India, or European. Or mixed. 

In table 1 I seen you have used skin colour. And Indigenous. Mixed could also be important.

In table 2 I miss the disabilities and the delay in diagnosis. These are very important to plan your extra focus points. And are more important than prevalence. And may help you to understand the epidemiology in that area.

280: Look not only at skin colour but at poverty too, that may be more important. I see you mention it in 285-288 and thereafter.

I wonder whether this specific research will contribute to the knowledge of the reader. Because all was known already.

Reviewer #5: Minor Revision

**Summary and General Comments**

Reviewer #2: The addition of the distribution of cases over the 2011-2020 timeframe in Santarem by GIS mapping and calculation of relative risk (RR) in neighborhoods using spatial epidemiological tools was an important addition to the paper and greatly improved the quality of the data presented.

Reviewer #4: Look at the delay at diagnosis and the reasons, Look at disability by age.

Reviewer #5: This study adequately describes epidemiological, clinical and geographical characterization of leprosy in this endemic are of Brazil. Some minor revisions are necessary.

PLOS authors have the option to publish the peer review history of their article (what does this mean?). If published, this will include your full peer review and any attached files.

Reviewer #2: Yes: John S. Spencer

Reviewer #4: No

Reviewer #5: Yes: MARIA PENA

Figure Files:

Data Requirements:

Please note that, as a condition of publication, PLOS' data policy requires that you make available all data used to draw the conclusions outlined in your manuscript. Data must be deposited in an appropriate repository, included within the body of the manuscript, or uploaded as supporting information. This includes all numerical values that were used to generate graphs, histograms etc.. For an example see here: http://www.plosbiology.org/article/info:doi%2F10.1371%2Fjournal.pbio.1001908#s5.
---

## [Editor Report · Decision Letter 3]

27 Feb 2024

Dear Dr. Valentim,

Thank you very much for submitting your manuscript "Epidemiological, Clinical, and Geographical Characterization of Leprosy in the County of Santarém-Pará: Insights for Effective Control and Targeted Intervention" for consideration at PLOS Neglected Tropical Diseases. As with all papers reviewed by the journal, your manuscript was reviewed by members of the editorial board and by several independent reviewers. The reviewers appreciated the attention to an important topic. Based on the reviews, we are likely to accept this manuscript for publication, providing that you modify the manuscript according to the review recommendations. 

The reviewers have patiently pointed out the changes that need to be made. In your response, you say that you have made the changes but when I check the manuscript, I see that there was no change made!

Please carefully address each issue. For example, "brown race" is still in the Abstract. Table 2 and the text still refer to Virchowian rather than lepromatous leprosy, the term that is most familiar to readers internationally. I expect that there remain other unaddressed changes. I have indicated minor revisions trusting that you will be able to make all the revisions.

Please revise the text thoroughly. If this is not possible, you should withdraw the manuscript and submit it elsewhere. Please make all the changes requested by the reviewers or explain why you disagree with the change.

Sincerely,

Paul J. Converse

Academic Editor

Ana LTO Nascimento

Section Editor

Dear Dr. Aguiar Valentim,

The reviewers have patiently pointed out the changes that need to be made. In your response, you say that you have made the changes but when I check the manuscript, I see that there was no change made!

Please carefully address each issue. For example, "brown race" is still in the Abstract. Table 2 and the text still refer to Virchowian rather than lepromatous leprosy, the term that is most familiar to readers internationally. I expect that there remain other unaddressed changes. I have indicated minor revisions trusting that you will be able to make all the revisions.

Please revise the text thoroughly. If this is not possible, you should withdraw the manuscript and submit it elsewhere. Please make all the changes requested by the reviewers or explain why you disagree with the change.

Figure Files:

Data Requirements:

Please note that, as a condition of publication, PLOS' data policy requires that you make available all data used to draw the conclusions outlined in your manuscript. Data must be deposited in an appropriate repository, included within the body of the manuscript, or uploaded as supporting information. This includes all numerical values that were used to generate graphs, histograms etc.. For an example see here: http://www.plosbiology.org/article/info:doi%2F10.1371%2Fjournal.pbio.1001908#s5.

Reproducibility:

References

---

## [Editor Report · Decision Letter 4]

11 Mar 2024

Dear Dr. Valentim,

We are pleased to inform you that your manuscript 'Epidemiological, Clinical, and Geographical Characterization of Leprosy in the County of Santarém-Pará: Insights for Effective Control and Targeted Intervention' has been provisionally accepted for publication in PLOS Neglected Tropical Diseases.

Best regards,

Paul J. Converse

Academic Editor

Ana LTO Nascimento

Section Editor

---

## [Editor Report · Acceptance letter]

18 Mar 2024

Dear Ph.D Valentim,

We are delighted to inform you that your manuscript, "Epidemiological, Clinical, and Geographical Characterization of Leprosy in the County of Santarém-Pará: Insights for Effective Control and Targeted Intervention," has been formally accepted for publication in PLOS Neglected Tropical Diseases.

Best regards,

Shaden Kamhawi

co-Editor-in-Chief

Paul Brindley

co-Editor-in-Chief
